# Exploring the Genetic Predisposition to Epigenetic Changes in Alzheimer’s Disease

**DOI:** 10.3390/ijms24097955

**Published:** 2023-04-27

**Authors:** Leonid O. Bryzgalov, Elena E. Korbolina, Tatiana I. Merkulova

**Affiliations:** 1The Federal Research Center Institute of Cytology and Genetics, The Siberian Branch of the Russian Academy of Science, 10 Lavrentyeva Prospekt, 630090 Novosibirsk, Russia; 2Vector-Best, 630117 Novosibirsk, Russia

**Keywords:** Alzheimer’s disease, epigenetics, histone modification, chromatin remodeling, non-coding regulatory SNPs, transcription factors, gene expression regulation

## Abstract

Alzheimer’s disease (AD) is a prevalent type of dementia in elderly populations with a significant genetic component. The accumulating evidence suggests that AD involves a reconfiguration of the epigenetic landscape, including DNA methylation, post-translational modification of histone proteins, and chromatin remodeling. Along with environmental factors, individual specific genetic features play a considerable role in the formation of epigenetic architecture. In this study, we attempt to identify the non-coding regulatory SNPs (rSNPs) able to affect the epigenetic mechanisms in AD. To this end, the multi-omics approach is used. The GEO (Gene Expression Omnibus) available data (GSE153875) for AD patients and controls are integrated to reveal the rSNPs that display allele-specific features in both ChIP-seq profiles of four histone modifications and RNA-seq. Furthermore, we analyze the presence of rSNPs in the promoters of genes reported to be differentially expressed between AD and the normal brain (AD-related genes) and involved in epigenetic regulation according to the EpiFactors database. We also searched for the rSNPs in the promoters of the genes coding for transcription regulators of the identified AD-related genes. These regulators were selected based on the corresponding ChIP-seq peaks (ENCODE) in the promoter regions of these genes. Finally, we formed a panel of rSNPs localized to the promoters of genes that contribute to the epigenetic landscape in AD and, thus, to the genetic predisposition for this disease.

## 1. Introduction

Alzheimer’s disease (AD) is currently ranked as the leading cause of dementia and disability in older people throughout the world. In the context of AD, an affected individual is facing a progressive decline in cognitive functions and memory, a wide range of behavioral problems, brain atrophy, and ultimately death [1]. Hallmark pathologies in the brains of AD patients include extracellular β-amyloid plaques and intracellular neurofibrillary tangles composed of abnormally highly phosphorylated forms of the tau protein [2], as well as abnormal accumulation of lipids [3]. These changes are associated with widespread neuronal cell death [4].

The genetics of AD supports a dichotomous model based on the age when symptoms first appear, which is generally accepted as 65 years old [5]. The highly penetrant mutations in the genes coding for amyloid precursor protein (APP), presenilin 1 (PSEN1), and presenilin 2 (PSEN2) are considered responsible for an earlier onset (early-onset, EOAD) or familial form, accounting for 5 to 10% of AD cases according to different estimates [6]. Approximately 90% of past clinical trials were performed in the late-onset (LOAD) or sporadic AD cases. The likelihood of LOAD development is linked to the interplay between a number of susceptibility genes, with the apolipoprotein E (*APOE*) gene as a main player, as well as environmental factors [7]. There are three common APOE allelic forms identified in humans—ε2, ε3, and ε4 [7]—which result from the haplotypes defined by two common SNPs: rs429358 (T/C) and rs7412 (C/T) [8]. APOE ε4 has been identified as a genetic risk factor for AD with dose-dependent effects [9] and is associated with an increased risk for cerebrovascular disease and ischemic stroke [10,11]. APOE alleles ε3 and ε2 are associated with neutral or protective roles in AD, respectively [12].

The years of GWAS efforts and, later, meta-analyses of GWAS identified over 30 independent replicable LOAD genetic risk variants in addition to the *APOE* alleles [13,14,15]. For instance, the associations for the SNPs in or close to *CR1*, *BIN1*, *MEF2C*, *PICALM*, *SORL1*, *CLU*, *ABCA7*, and *CD33* genes were described. A comparable amount of 42 novel risk loci was reported in 2022 as a result of the two-stage study comprising 111,326 clinically diagnosed AD cases and 677,663 controls [16]. Generally, the most significant gene sets associated with LOAD are related to the amyloid and tau metabolism, brain lipid metabolism, and immune processes, including macrophage and microglial molecular networks [17,18,19].

Among the AD-associated variants are those affecting protein structures, including the rare coding variants. For example, the AD phenotype is regulated via signaling of TREM2, a microglial receptor that interacts with Aβ oligomers, facilitates amyloid clearance, and contributes to the APOE-mediated regulation of immune processes [20]. However, the bulk of AD-associated SNPs are non-coding and considered to mediate their effects via altered gene expression, which is likely to occur in a cell type-dependent manner [21,22].

Recently, an extensive research has suggested an important role for the epigenetic mechanisms, including DNA methylation, histone posttranslational modification, and non-coding RNA regulation, in the course and development of AD [23,24,25,26,27]. Typically, characteristic of the affected brain are abnormalities in DNA methylation and histone modification patterns that alter the gene expression at a transcriptional level via upregulation, downregulation, or silencing [28].

For instance, the AD onset and progression is attributed to the changes in genome-wide patterns of histone acetylation in post mortem brain samples [29]. Acetylation dysregulation in the human AD brain is associated with various impairments in signaling, proliferation, inflammation, immunity, apoptosis, and neuronal plasticity [30]. Moreover, in AD patients higher levels of histone deacetylases (HDACs), which are the enzymes that catalyze the removal of acetyl groups leading to transcription repression, have been observed in certain brain regions responsible for learning, memory, and neuroplasticity. The increases in HDACs are associated with the impairments in cognitive and synaptic functions [31]. Histone methylation-related epigenetic defects in AD include the upregulation of a number of histone methyltransferases and histone demethylases [32,33], in particular a number of H3K4me3 methyltransferases. Correspondingly, the H3K4me3 related to expression activation is often reported significantly increased in the AD brain, for example in the pre-frontal cortex of AD patients and P301S human tau transgenic mice [29]. The H3K4 methylation status contributes to the processes of signal transmission in synapses [34] and may promote the expression of memory-related genes and proteins, such as *ZIF268* and BDNF [35,36]. A marker of gene silencing—increased trimethylation of lysine 9 in histone H3 (H3K9) [37]—as well as higher levels of histone methyltransferase EHMT1 mRNA were also observed in the post mortem brain of AD subjects [38].

Other histone modifications (e.g., phosphorylation, ubiquitination, or SUMOylation) may modulate the transcription machinery binding with the chromatin and, as a result, add to AD pathogenesis [39,40] The corresponding protein machinery that recognizes, adds, or removes the chemical tags (i.e., acetyl, methyl or phosphate groups) within histone tails is often classified as histone modification readers, writers, and erasers with respect to the function [41]. The ATP-dependent chromatin remodeling factors are also pivotal regulators of gene expression in synaptic plasticity and memory formation [42,43] and in age-related neurodegenerative disorders, including AD [44]. For example, a critical involvement was shown for nBAF complex from the SWI/SNF sub-family [45].

Thus, a growing body of evidence suggests that a widespread reconfiguration of the epigenome in the course of AD is closely involved in the disease etiology [35]. Both individual specific genetic features and environmental factors underlie the observed epigenetic changes [46]. In this study, we have analyzed the multiomics data for post mortem human AD brain samples and controls [32], aiming to identify the non-coding regulatory variation (regulatory SNPs, rSNPs) putatively affecting the epigenetic mechanisms in AD and leading to changes in gene expression.

## 2. Results

### 2.1. Discovering the rSNPs Associated with Allele-Specific Events

Given the publicly available sequencing data, we have identified 1563 rSNPs associated with both allele-specific binding and expression within ±1000 bp from the TSSs (transcription start sites) or, in total, 1286 known human genes, as well as representatives for the promoter regions. Appendix A outlines the complete list. We noted that some rSNPs are linked to the allele specific expression of more than one target gene; the variants that were not assigned an identification tag via NCBI were excluded from further analysis. The input was H3K4me3, H3K4me1, H3K27ac, and H3K9ac profiling data (ChIP-seq) and transcriptomes (RNA-seq) obtained with the same samples from the lateral temporal lobe of 12 AD patients and 18 controls of matched age by Nativio et al. (see Section 4 for details). Figure 1 shows the main stages in the used bioinformatics algorithm.

### 2.2. Identifying the rSNPs Affecting the Expression of DEGs (Differentially Expressed Genes) between the Temporal Lobe of AD Patients and Controls

To find out the role of the discovered rSNPs in transcriptional regulation in AD, we examined the promoter regions of the genes differentially expressed in the lateral temporal lobe between AD patients and age-matched controls (GSE153873). The analyzed list comprised 421 genes with significant upregulation and 434 genes with significant downregulation in AD. These genes are referred to as AD-related genes because their abnormal expression was considered putatively involved in the AD pathogenesis [32]. We further searched for the rSNPs that might be responsible for the observed changes in expression. As a result, we identified 42 rSNPs in the promoters of 36 upregulated AD-related genes and 18 rSNPs in the promoters of 15 downregulated AD-related genes (Appendix A). The protein–protein interaction (PPI) network was further built with STRING for these 51 targets (Figure 2). The most representative enriched gene ontology (GO) term turned out to be ‘chromatin remodeling’ (Appendix A).

According to EpiFactors 2.0 as of 7 July 2022 [47], seven upregulated genes—*ARID1B*, *BANP*, *DAXX*, *HDAC4*, *SIRT1*, *TRRAP*, and *UHRF1*—are well-characterized epigenetic regulators (Table 1). In contrast, none of the downregulated AD-related genes were annotated in the database in relation to epigenetic processes.

The encoded proteins are classified in three main categories related to histone modification: histone modification writing, reading, and erasing (though some were involved in more than one functional type). Three proteins (SIRT1, TRRAP, and UHRF1) act as the co-factors forming complexes with epigenetic factors involved in histone modification writing; four proteins (ARID1B, UHRF1, HDAC4, and SIRT1) are described in the literature as active enzymes.

*ARID1B* is one of the most frequently (~1%) mutated genes with its loss-of-function variants associated with intellectual disability [49]. The encoded E3 ubiquitin ligase is a component of well-studied eukaryotic chromatin-remodeling SWI/SNF-A type complex, targeting SWI/SNF to specific genes [50] and activating or repressing the expression. ARID1B is most likely involved in the AD pathogenesis because it plays a certain role in cell cycle activation [51]. Another gene from the list coding for type E3 ubiquitin ligase is *UHRF1*. Nishiyama et al. [52] first demonstrated that UHRF1 could ubiquitinate histone H3 at K23; the modified H3 was subsequently recognized via the maintenance DNA methyltransferase DNMT1 [53]. UHRF1 also regulates DNMT1 via proper genome targeting via binding to either H3K9me2/3 or hemi-methylated CpG regions [54]. Interestingly, DNMT1 methyltransferase and UHRF1 as its co-factor are the primal players in maintenance of imprinted methylation during the early development in mammals [55,56]. *SIRT1* codes for NAD+ dependent histone deacetylase [57], which is an epigenetic modifier involved in histone acetylation and methylation, and targets different histone substrates. Thus, SIRT1 de-acetylates several lysines at the positions related to transcription silencing: H1K26, H4K16, H3K9, and H3K14 [58]. The pivotal role of SIRT1 was widely established in cellular senescence and neurodegenerative diseases, including AD [59]. Thus, SIRT1 is able to bind HDAC1 deacetylase, which contributes to the reduction in DNA damage in mice [60]. As has been shown, the level of another histone deacetylase—HDAC4—in the cell nuclei is considerably increased in the brain of AD human subjects and in AD mouse models [61,62] in connection with abnormal synaptic function. Furthermore, the overexpression or over-representation of HDAC4 in the cytoplasm can recover synaptic transmission [62,63]. There is also evidence that HDAC inhibitor treatments restore brain damage and cognitive performance in AD mice [64,65,66]. The protein product of *TRRAP* is the only member in the phosphatidylinositol 3-kinase-related kinase (PIKK) family that lacks enzymatic activity. TRRAP is an essential component of many histone acetyltransferase (HAT) complexes, which recruits the complexes to chromatin acting in transcription and DNA repair [67]. BANP is described as a protein involved in histone H3 and H4 acetylation. It binds to unmethylated CGCG elements and, thus, is a critical regulator in the expression of CpG-island–regulated genes [68]. DAXX protein interacts directly with H3.3 histone core [69]. The findings suggest that DAXX acts as specific chaperone that deposits H3.3 to maintain the H3K9me3 modification throughout the genome in the context of ATRX/DAXX complex [70]. A certain role of DAXX in the prevention of amyloid fibrillization and plaque build-up was recently shown [71].

### 2.3. Identifying the rSNPs That Affect the Expression of Known Epigenetic Regulators

Furthermore, we revealed rSNPs within the promoter regions of all genes coding for epigenetic regulators based on the EpiFactors datasets [47]. Of the 1563 rSNPs, we found 132 rSNPs in 97 target genes related to epigenetic processes, including chromatin remodeling (*n* = 34), histone modification writing (*n* = 90), histone modification reading (*n* = 48), and histone modification erasing (*n* = 44). Hereafter, these genes are referred to as ‘Epi’ genes (Appendix A).

None of these 132 rSNPs were listed in any GWAS related to AD [72]. Notably, none of the 51 rSNPs discovered in the promoters of AD-related genes have entered the GWAS catalog. Presumably, this issue results from an extremely low SNP heritability of AD, as reported recently [73]. When annotated through the Disease Ontology (DO) database [74], one gene out of 97—*HMGB1* (rs1020625837)—was classified as related to AD-caused dementia. The cumulative evidence highlights the contribution of HMGB1 and its receptor signaling in neuroinflammation [75,76] with the beneficial effects on therapeutic targeting to delay the AD onset.

Nevertheless, five of the 97 epigenetic regulators (about 5%) were previously reported in the literature in relation to AD. For example, KANSL1 protein influences gene expression involved in histone H4 lysine 16 (H4K16) acetylation [77]. The locus of *KANSL1* gene in chromosome 17 near *MAPT*, which was coding for the tau protein, was identified as an AD risk locus in the International Genomics of Alzheimer’s Project (IGAP) based on *APOE* individual status [78]. Examination of human datasets shows that the expression of two additional Epi genes—*BAZ2B* (encoding bromodomain protein) and *EHMT1* (encoding histone methyltransferase)—in the frontal cortex increases with age and positively correlates with AD progression [79]. The roles of several proteins in promising therapeutic strategies for AD were also reported. In particular, it is shown that the overexpression of *CTBP1* confers the attenuation of apoptosis of hippocampal and cortical neurons in different AD rat models [80]. CTBP1 displays both the co-repressor and co-activator functions acting through direct interactions with a variety of epigenetic regulators or their multicomponent complexes, which include p300 protein, histone deacetylase, histone methyltransferases, and a brain-specific component of the SWI/SNF complex known as ArpNα [81,82]. SIRT1 was already mentioned here as an epigenetic regulator performing important functions in neurodegenerative diseases, including AD, with an rSNP (rs1053224730) found in its promoter. In addition, SIRT1 regulates the non-amyloidogenic processing of APP [59] and protects against microglia-dependent Aβ toxicity in vitro via inhibiting NF-κB signaling [83]. Moreover, strong correlations between SIRT1 levels and the cognitive skills in AD patients were illustrated: the individuals with a higher *SIRT1* expression displayed significantly better cognitive functioning [84].

Thus, we executed the first study to form a list of the rSNPs in the promoters of genes coding for epigenetic regulators putatively related to AD susceptibility and progression.

### 2.4. Identifying the rSNPs in the Promoters of Transcription Factors Involved in the Regulation of AD-Related Genes

For a more comprehensive picture of the rSNPs involvement in the epigenetic mechanisms of AD, we attempted the search for such SNPs in promoters of genes coding for transcription factors and other regulatory proteins providing the transcriptional control of AD-related genes.

With this in mind, we first overlapped the promoters of 855 AD-related genes with ENCODE 3 ChIP-seq tracks available for 340 DNA and chromatin binding proteins in 129 cell types in hg38 as of October 2022 [85]. Some proteins contained a DNA binding domain and could directly bind to specific short DNA motifs (mainly transcription factors); others bound to DNA indirectly through interactions with DNA-bound transcription factors. If an ENCODE ChIP-seq peak for specific (ENCODE-derived) protein was located within the promoter of AD-related gene, we checked whether any rSNP was located within the promoter of the gene coding for this protein (Figure 3).

As a result, we found that 49 of 1500 rSNPs (Appendix A) were located within the promoter regions of 37 genes whose protein products were tracked in promoters of 10 AD-related genes using ENCODE 3. The number of targets for one ENCODE-derived protein between two and 10; thus, a complex transcriptional regulation of these 10 AD-related genes is evident. Appendix A gives the full list of identified interactions.

As expected, the list of ENCODE-derived proteins that bind to the promoters of the genes deregulated in AD [32] contains a significant share of the well-known transcription factors (Table 2). Accordingly, our findings suggest the involvement of all these transcription factors in AD etiology.

This statement is partially confirmed by comparison the data currently available in the relevant literature. Thus, *MEF2A* highly expressed in microglia, as shown via a single-cell sequencing [86], is found significantly downregulated in the AD brain in strong correlation with the expression of seven autophagy-related genes [87], suggesting that MEF2A may be closely associated with the AD pathogenesis, thereby inhibiting autophagy. In addition, the allele-specific binding of some transcription factors, including NFIB, NFIC, and CUX1, to the functional SNPs within *HLA-DRB1/DQA1* locus associated with a LOAD onset via GWAS was reported. The authors have shown that these proteins do regulate the expression of both HLA-DQA1 and HLA-DRB1 in microglia [88]. In addition, AD phenotype–genotype correlations were recently demonstrated for *TCF12* [89]. However, apart from the regulatory proteins which have been already established in AD, we propose some novel candidates which merit a careful study.

Most interestingly, among 37 ENCODE-derived proteins there were 12 epigenetic regulators, including both the protein previously described here (ARID1B, Table 1) and the ones newly linked to AD (Table 3).

For the 11 newly identified epigenetic regulators, three proteins were related to histone modification writing: CLOCK, EHMT2, and KAT8. In addition, MGA, described as a transcription factor with both TBOX and bHLH zip DNA-binding domains at once [90], was classified as a histone modification writing co-factor in EpiFactors. Two proteins (RCOR1 and SAP30) were reported as histone modification erasing co-factors both involved in modification of the histone acetylation profile. We also identified five proteins described as chromatin remodelers: the components of well-studied NuRD (CHD4) and SWI/SNF (DPF2) epigenetic complexes; chromatin remodeling co-factors (NCOA2 and SMARCE1); and NADH-dependent nuclear regulator CTBP1, acting through the interactions with a variety of epigenetic regulators.

Summing up, in Section 2.2 we gave a description of seven epigenetic regulators which are harboring rSNPs in their promoters, deregulated in AD, and likely involved in pathogenic mechanisms. Here we identified 11 more genes (with a total of 15 more rSNPs in their promoters) that may add to epigenetic panorama in AD. The data suggest the crosstalk between the genes (and their protein products) deregulated in the temporal lobe of AD subjects with the applicability to AD pathogenesis. Thus, *ARID1B*, a component of SWI/SNF chromatin-remodeling complex, was reportedly a plausible candidate with the transcription upregulated in the temporal lobe in AD subjects [32], presumably because of the allele-specific effects of the rs2281391 variant, which we identified in the *ARID1B* promoter region (Figure 4a). The ChIP-seq peaks for the ARID1B protein are located within the promoters of another six AD-related genes, including the genes coding for well-described transcription factors (DP2 and ZBED6) and epigenetic regulator DAXX (Figure 4b). The expression of five of the six ARID1B targets was upregulated in the AD subjects when compared to controls (Figure 4c). As is known, SWI/SNF complexes generate and maintain the chromatin accessibility through loosening the histone-to-chromatin binding [91]. Accordingly, the presence of ARID1B in the promoter regions of target genes agrees with the reported upregulation of *ZBED*, *ZC3H11A*, *AGAP1*, *TFDP2*, and *DAXX*. In contrast, the *HSD17B8* expression was downregulated in the temporal lobe in AD, which suggests a more intricate network of transcription regulation. Correspondingly, we have identified rSNPs that may also contribute to the transcription profiles in the promoters of all ARID1B targets (Figure 4b).

## 3. Discussion

Emerging studies have shown that the multilayered profiles of epigenetic dysregulation, such as DNA methylation, histone modifications, chromatin remodeling, and non-coding RNA regulation, are intimately involved in various complex disease pathogenesis [92,93,94,95], including neurological diseases [31,96,97,98]. It is known that along with the impact of environmental factors, genetic pre-disposition plays an important role in the epigenetic changes associated with various pathologies [39,46]. Numerous examples of different pathologies related to genetic variation (SNPs) underlying the abnormalities in the expression or function of the genes coding for epigenetic factors, including histone acetylases [99,100], deacetylases [101,102,103], histone methyltransferases [102,104], and components of chromatin remodeler complexes [105,106], have been thus far accumulated. However, most of these results have been obtained when studying different types of cancer. For example, two SNPs (rs6950683 and re3757441) of *EZH2* coding for the histone methyltransferase that causes the trimethylation of H1K26 and, consequently, suppresses the cancer preventive genes were shown to be associated with a tumor size in triple-negative breast cancer [102]. Further, the association-based study of SNP rs201135441C>T shows that the T allele of rs201135441 significantly increases the risk of breast cancer susceptibility and reduces the overall survival rate [107]. One of the 65 new breast cancer risk loci identified via GWAS—rs4971059—is shown to activate the expression of ubiquitin ligase TRIM46, which targets histone deacetylase HDAC1 for ubiquitination and degradation [101]. Another SNP—rs4903064—confers an allele-specific effect on the expression of DPF3, which is a component of the BAF complex and part of the SWI/SNF complexes. An allele-specific overexpression of DPF3 in renal cell lines can lead to reduced apoptosis and activation of the STAT3 pathway, which are both critical in RCC carcinogenesis [106].

In contrast, the searches for SNPs able to influence the expression of function in the genes coding for epigenetic regulators and concurrently associated with neurodegenerative diseases, especially AD, are almost absent. On the other hand, the emerging paradigms demonstrate that dynamic and latent epigenetic alterations are widely incorporated into the AD pathological pathways [25,39,108]. Thus, the goal of our work was to detect the SNPs able to affect the epigenetic mechanisms in AD. As it is known that the majority of the SNPs (up to 90%) associated with the trait reside in the regulatory region of the genome (promoters, enhancers, and so on) and influence gene expression [109,110,111], we focused on the search for this particular regulatory SNPs (rSNPs) using our own earlier developed functional approaches based on the search for allelic specificity of events in multi-omics data [108,109,112].

We started with the sequencing data from the post mortem human AD brain samples (temporal lobe) and age-matched control brain samples, including the ChIP-seq profiling for H3K4me3, H3K4me1, H3K27ac, and H3K9ac and the RNA-seq transcriptome profiling (GSE153875). The sample size was 30, including 12 samples from the AD group and 18 controls from the matching age. We noted that evidence exists suggesting that “pure” pathologies may be rare and most subjects are likely to have a mix of more than one type of dementia [113]. To this end, our study involved subjects that were thoroughly phenotyped using clinical and neuroimaging data [32].

Here, the raw data (GSE153875) for AD patients and controls were integrated to reveal the rSNPs associated with the allele-specific events in both ChIP-seq profiling of four histone activating modifications (allele-specific binding, ASB) and RNA-seq (allele-specific expression, ASE). An important specific feature when using ASB and ASE for the detection of potential rSNPs rather than the widely applied eQLT analysis and, more so, GWAS to assess the SNPs of numerous individuals in the background of quite different genomic content and life conditions, consists of the fact that the allele-specific events are recorded for each individual on the same background [111]. This feature makes it possible to obtain reliable data when studying either a very small number of individuals or a single person [114]. An increase in the sample size is necessary for involving a larger number of the SNPs in a heterozygous state. In other words, the studies on a larger number of people will enable mapping of more potentially regulatory variants. In particular, the performed computations demonstrate that the data for 20 individuals theoretically allows the allele-specific events to be determined for 65–70% of the SNPs with a population frequency of ≥5% [115]. However, some conclusions of our study were limited by technical aspects of the experiment’s design. As we were constrained by the available sequencing data, the number of identified allele-specific events depend on the presence of individuals heterozygous by specific positions in the sample. Moreover, our approach for predicting allele-specific effects assumes that the effects are significant, which suggests that the analyzed heterozygous positions are sufficiently covered in ChIP- and/or RNA-seq data. Therefore, we did not analyze many low-expressed genes. Thus, using the described data, we have identified approximately 1500 rSNPs (in fact, 1563) associated with both allele-specific binding and expression (Appendix A) within ±1000 bp from the TSSs of 1286 genes. This analysis of genes with expression altered in the brain samples of AD human subjects [32] allowed us to find 60 rSNPs in the promoters of 51 genes reported to be deregulated in the temporal lobe in AD, including 36 upregulated AD-related genes and 15 downregulated AD-related genes (Appendix A).

Remarkably, we found the target genes being generally enriched in the ‘Chromatin Remodeling’ GO category (Figure 2). According to EpiFactors 2.0, seven upregulated genes—*ARID1B*, *BANP*, *DAXX*, *HDAC4*, *SIRT1*, *TRRA*, and *UHRF1*—were identified as well as characterized epigenetic regulators (Table 1). The corresponding protein products included type E3 ubiquitin ligases (ARID1B and UHRF1), histone deacetylases (SIRT1 and HDAC4), and an essential component of histone acetyltransferase complexes (TRAPP). These findings are in line with experimental data that highlight the emerging molecular mechanisms in AD as well as the future strategies aiming to exploit the ubiquitin system and histone acetylation/deacetylation as sources of next-generation therapeutics [30,116,117,118,119].

In addition, we attempted to search for the rSNPs within the promoter regions of all genes coding for epigenetic regulators referred to in EpiFactors datasets and discovered 132 SNPs of this type in 97 target genes. However, only six of these genes emerged to be associated with AD in a certain way: *HMGB1*, which is related AD dementia according to ‘Disease Ontology’; *KANSL1*; *BAZ2B*; *EHMT1*; *SIRT1*; and *CTBP1*, which are associated with AD according to the literature data [55,73,75,76]. This result is not unexpected since the sample of 1500 rSNPs, which we formed based on the analysis of allele-specific events in the lateral temporal lobe (without distinguishing the DEGs associated with AD), undoubtedly contains the variants associated with many brain-related traits, including the predisposition to other neuropathologies. In addition, such analysis can also detect a number of rSNPs, the effects of which are implemented in other tissues.

The search for the rSNPs in promoters of transcription regulators for the distinguished group of AD-related genes was more effective. This approach allowed us to detect approximately 50 rSNPs in the promoters of 37 genes involved in the transcriptional regulation in AD. Most of them (*n* = 25) were known transcription factors (Table 2) and epigenetic regulators (Table 3). In particular, the latter comprised 11 proteins, including those involved in histone methylation (EHMT2 and MGA) and acetylation (CLOCK, KAT8, MGA, RCOR1, and SAP30), as well as the components of NuRD and SWI/SNF epigenetic complexes. Thus, the list of discovered rSNPs involved in the formation of the epigenetic landscape in AD was reported.

## 4. Materials and Methods

### 4.1. Brain Epigenomic and Transcriptomic Data

This study was conducted based on the raw experimental data originally published by Nativio et al. [32], which were downloaded from the NCBI Gene Expression Omnibus (GEO accession number GSE153875). ChIP-seq profiling of histone modifications (H3K4me3, H3K4me1, H3K27ac, and H3K9) and expression profiles via RNA-seq were available for the same post mortem human brain samples collected from the lateral temporal lobe of the patients neuropathologically diagnosed with AD (but not any similar neurodegenerative disease, *n* = 12) and cognitively healthy controls (two age groups, *n =* 18); the patients were mainly male subjects. Informed consent for autopsy was obtained for all patients; consent was approved as defined in the original publication.

### 4.2. Human Genome Data

We used the NCBI GRCh38/hg38 (Genome Reference Consortium Human Build 38) reference genome available through the NCBI FTP site [120] with TSSs (transcription start sites) information. The initial list of human transcripts was downloaded from the UCSC data portal.

### 4.3. Sequencing Data Preprocessing and Alignment

The raw data were initially processed as described by Korbolina et al. [121]. Briefly, the reads were quality-filtered according to the Illumina pipeline (phred ≥ 20, *n* ≥ 32) and the adapters were cut off using Trimmomatic v. 3.2.2. The data were aligned to GRCh38 human reference genome with Bowtie2 (version 2.2.3). Furthermore, we analyzed only the SNPs that were indexed in the dbSNP Database for single nucleotide polymorphisms [48]. The individual alternative reference genome sequence replacing the reference bases in polymorphic positions with the bases representing the alternative alleles was built for each examined person. In order to increase the coverage, ChIP-seq and RNA-seq reads from all files of one human brain sample were pooled. The realignment step to both GRCh38 and appropriate alternative reference finished the pre-processing step. Except for the mentioned steps, all steps were performed within R/Bioconductor environment.

### 4.4. Assessing Allele-Specific Events

A bias analysis was performed as earlier described [121]. In short, we identified the allele-specific expression (ASE) or allele-specific binding (ASB) events if the number of reads that fall to the reference allele were enriched or depleted in a statistically significant manner, considering the correction for multiple comparisons (odds ratio ≥ 1.5; padj ≤ 0.1). Two consecutive binomial tests were applied (as visualized on Figure 1). All steps were performed within R/Bioconductor environment unless otherwise stated.

### 4.5. Assignment of Gene Promoter Regions

The promoter regions were analyzed within 1 Mb windows in either direction from the TSSs.

### 4.6. Deposited Data

#### 4.6.1. The Encyclopedia of DNA Elements (ENCODE)

Profiling of the ChIP-seq peaks was based on ENCODE 3 Transcription Factor ChIP-seq Peaks track via R.

#### 4.6.2. GWAS Catalog

We examined GWAS index SNPs available up to October 2022. We used the same ‘AD’ signature for querying the GWAS Catalog as for querying PubMed.

#### 4.6.3. The EpiFactors Database

A catalog of human proteins involved in epigenetic processes (epigenetic regulators), their complexes, the corresponding genes, and targets was found in the EpiFactors 2.0 Database [47].

#### 4.6.4. PubMed Resource

Research papers and reviews published from 2010 to 2023 were identified through PubMed [122] with keywords including rSNP IDs as in dbSNP, targeted genes’ official symbols, and the concepts ‘Alzheimer’s’, ‘early-onset AD’, ‘late-onset AD’, and ‘familial history of AD’.

### 4.7. p-Value Correction

Benjamini–Hochberg correction was used to adjust the raw *p*-value; separate procedures were performed for RNA-seq and ChIP-seq data.

## 5. Conclusions

Overall, it is clear from the literature and extensive research thus far that various epigenetic malfunctions are intimately involved in the course of AD [25,28,33,35]. All epigenetic changes that can contribute to the AD onset and progression result from the interplay of the environmental cues and genome. Some changes occur long before the molecular pathology of AD develops and, thus, have been highlighted as promising targets for AD diagnostic or therapeutics. Thus, the epigenetic events may be more upstream in the AD pathology than the more common or conventional clinical features, such as BACE, γ-secretase, Aβ, and tau [123]; however, an in-depth analysis of the involved mechanisms is highly desired.

## Figures and Tables

**Figure 1 ijms-24-07955-f001:**
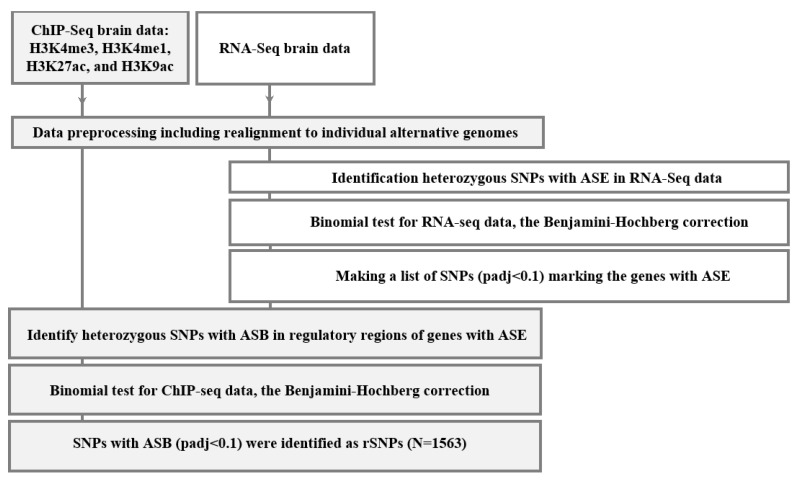
Overall computational algorithm to search for rSNPs. ASE is allele-specific expression; ASB, allele-specific binding; and padj, *p*-value corrected using the Benjamini–Hochberg method.

**Figure 2 ijms-24-07955-f002:**
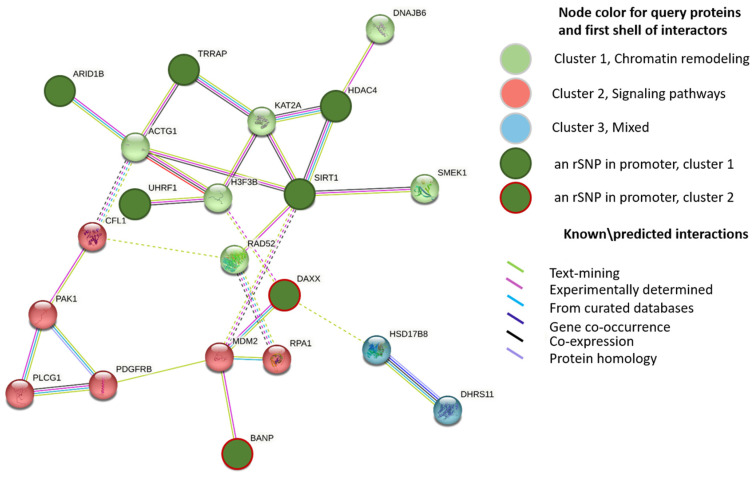
PPI STRING network for 51 DEGs with rSNPs in their promoters and first shell of interactors (k-means clustering). Names are given to clusters according to most enriched or common GO categories found. Dotted lines show the existing edges between clusters; disconnected nodes and proteins with a single interactor are hidden.

**Figure 3 ijms-24-07955-f003:**
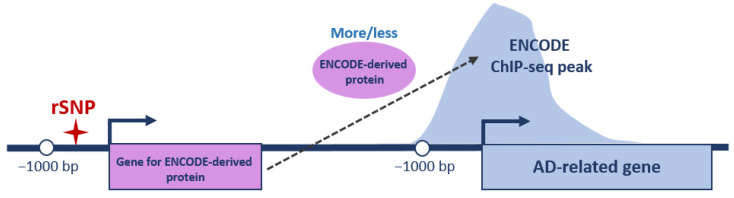
Schematic representation of analyzed regulatory interactions. ENCODE protein/ENCODE ChIP-seq peak—a DNA/chromatin binding protein tracked in ENCODE and corresponding peak from ChIP-seq profiling, respectively; ✦—rSNP position; bent arrows—positions of TSSs and direction of transcription; and bp—base pairs. Position −1000 bp restricts gene promoter region on left.

**Figure 4 ijms-24-07955-f004:**
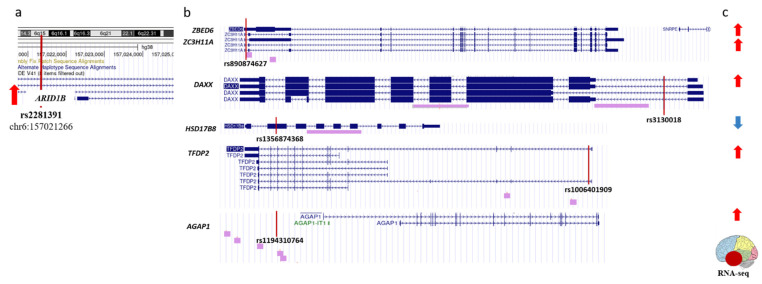
Schematic representation of regulatory interactions discovered for ARID1B. (**a**). Regulatory variant (rs2281391) is identified within the *ARID1B* promoter. (**b**). ARID1B binds to promoters of six target genes. Purple bars below sequence alignments from reference assembly indicate ChIP-seq peaks from ENCODE 3. (**c**). All six ARID1B targets are differentially expressed in temporal lobe of AD patients as compared with age-matched control brains. (**a**,**b**). Positions of rSNPs on panels are indicated with vertical red lines as tracked via UCGC genome browser on human GRch38/hg38. (**a**,**c**). Red/blue arrows shows up- and down-regulated transcription, respectively.

**Table 1 ijms-24-07955-t001:** Brief overview of epigenetic regulators upregulated in temporal lobe of AD subjects.

rSNP_ID	Gene Symbol	Function	Modification	PMID	Complex Name	Target	Specific Target
rs2281391	*ARID1B*	HMW	Histone ubiquitination	20086098	BAF, nBAF, npBAF, PBAF, SWI/SNF-like_EPAFa, SWI/SNF-like EPAFB, SWI/SNF BRM-BRG1	Histone, DNA	H2BK120, DNA motif
rs536224963	*BANP*	HMW	Histone acetylation	16166625	#	Histone	H3K9, H4K8
rs3130018	*DAXX*	#	#	23075851	#	Histone	H3.3
rs946554101	*HDAC4*	HME	Histone acetylation	10220385	#	Histone	H2AKac, H2BKac, H3Kac, H4Kac
rs1053224730	*SIRT1*	HME, HMW cofactor	Histone acetylation, Histone methylation	15469825	eNoSc	Histone	H1K26ac, H3K9ac, H4K16ac
rs1249650489	*TRRAP*	HMW cofactor	Histone acetylation	14966270	SWR, PCAF, TFTC-HAT, NuA4, SAGA, NuA4-related complex, STAGA	Histone	#
rs958341678	*UHRF1*	HMR, HMW cofactor	Histone ubiquitination	17967883	#	Histone, DNA	H3K9me3, H3R2, H3, mCG

rSNP_ID—identifier for an rSNP as in dbSNP [48]; HMW/HMR/HME—histone modifications writers/readers/erasers, respectively; complex name—the protein acts as a component of the designated protein complex if recognized in EpiFactors database; PMID—the identifier for PubMed search for an associated publication; and #—a lack of information in the database.

**Table 2 ijms-24-07955-t002:** Transcription factors involved in expression regulation of AD-related genes using ENCODE.

rSNP ID	TF	Target Genes	N
rs1022095596	CLOCK	*ZBED6*, *HSD17B8*, *ZC3H11A*, *SMG5*, *DAXX*, *GPC1*, *HDAC4*	7
rs977886453 rs1279727503 rs924734233	DPF2	*AGAP1*, *HERC6*, *DAXX*, *ZBED6*, *ZC3H11A*, *HSD17B8*, *HDAC4*, *TFDP2*, *SMG5*	9
rs1370216229	CUX1	*TFDP2*, *ZBED6*, *ZC3H11A*	3
rs1255551090	BCL11A	*HERC6*, *AGAP1*	2
rs995147107	FOSL2	*TFDP2*, *HDAC4*, *DAXX*, *AGAP1*, *HSD17B8*, *SMG5*	6
rs992579579	FOXK2	*DAXX*, *HSD17B8*, *HDAC4*, *TFDP2*, *AGAP1*, *ZC3H11A*, *SMG5*, *ZBED6*	8
rs1465639308 rs975045833	IRF1	*DAXX*, *ZC3H11A*, *HDAC4*, *TFDP2*, *HERC6*, *GPC1*, *AGAP1*, *SMG5*, *ZBED6*, *HSD17B8*	10
rs2570800 rs1402353341	MEF2A	*SMG5*, *HERC6*, *ZC3H11A*, *AGAP1*, *ZBED6*, *SMG5*, *HERC6*, *ZC3H11A*, *AGAP1*, *ZBED6*	10
rs1286079777	MGA	*HDAC4*, *TFDP2*, *AGAP1*, *ZC3H11A*, *ZBED6*, *DAXX*	6
rs954995579 rs1043408625 rs954995579	NFIB	*HERC6*, *ZBED6*, *DAXX*, *HDAC4*	4
rs983776002	NFIC	*SMG5*, *DAXX*, *ZBED6*, *ZC3H11A*, *HSD17B8*, *HERC6*, *AGAP1*, *TFDP2*, *HDAC4*	9
rs1484805397	SREBF1	*HDAC4*, *ZBED6*, *ZC3H11A*	3
rs1044184380	TCF12	*AGAP1*, *ZBED6*, *ZC3H11A*, *HSD17B8*, *HDAC4*, *HERC6*, *GPC1*, *SMG5*, *TFDP2*, *DAXX*	10

rSNP ID—a reference SNP identification number as in dbSNP; TF—official symbol for transcription factor tracked in ENCODE; target genes—official symbols for AD-related genes with a ChIP-seq peak for a specific protein in promoter; and N—the number of target genes for a certain TF.

**Table 3 ijms-24-07955-t003:** Overview of epigenetic regulators involved in the expression regulation of AD-related genes via EpiFactors.

rSNP ID	ENCODE-Derived Protein	Target Genes	Function	Modification	PMID	Complex Name	Specific Target
rs2281391	ARID1B	*AGAP1* *DAXX* *HSD17B8* *TFDP2* *ZBED6* *ZC3H11A*	HMW	Histone ubiquitination	20086098	BAF, nBAF, npBAF, PBAF, SWI/SNF-like_EPAFa, SWI/SNF-like EPAFB, SWI/SNF BRM-BRG1	H2BK120DNA motif
rs1363175143	CHD4	*DAXX* *HSD17B8*	CR	#	12592387	NuRD	#
rs1022095596	CLOCK	*DAXX* *GPC1* *HDAC4* *HSD17B8* *SMG5* *ZBED6* *ZC3H11A*	HMW	Histone acetylation	#	#	H3, H4
rs1013929495	CTBP1	*AGAP1* *DAXX* *GPC1* *HDAC4* *HERC6* *HSD17B8* *SMG5* *TFDP2* *ZBED6* *ZC3H11A*	CR	#	21102443	LSD-CoREST	#
rs977886453 rs1279727503 rs924734233	DPF2	*AGAP1* *DAXX* *HDAC4* *HERC6* *HSD17B8* *SMG5* *TFDP2* *ZBED6* *ZC3H11A*	CR	#	21888896	SWI/SNF BRM-BRG1	#
rs1465945079	EHMT2	*AGAP1* *DAXX* *HDAC4* *ZBED6*	HMW	Histone methylation	18264113	#	H3K9
rs777573795	KAT8	*ZBED6* *ZC3H11A*	HMW	Histone acetylation	10786633	NSL, CHD8, MLL2/3, COMPASS-like MLL1,2, MLL4/WBP7	H2A, H3, H4
rs1286079777	MGA	*AGAP1* *DAXX* *HDAC4* *TFDP2* *ZBED6* *ZC3H11A*	HMW cofactor, TF	Histone methylation, histone acetylation, TF activator, TF repressor	#	RING2-L3MBTL2, CHD8, MLL2/3, MLL4/WBP7	DNA motif
rs563166047	NCOA2	*ZC3H11A*	CR cofactor	#	9590696	#	#
rs1320061320 rs1395087048	RCOR1	*AGAP1* *DAXX* *GPC1* *HDAC4* *HERC6* *HSD17B8* *SMG5* *TFDP2*	HME cofactor	Histone acetylation,	10449787	BHC, SCL, LSD-CoREST	#
rs1346876773	SAP30	*DAXX* *GPC1* *HSD17B8* *SMG5* *TFDP2* *ZBED6* *ZC3H11A*	HME cofactor	Histone acetylation	9651585	mSin3A, mSin3A-like complex	#
rs930121077 rs907151175	SMARCE1	*AGAP1* *DAXX* *HSD17B8* *ZBED6* *ZC3H11A* *ZC3H11A*	CR cofactor	#	12672490	BAF, nBAF, npBAF, PBAF, SWI/SNF_Brg1(I), SWI/SNF_Brg1(II), SWI/SNF_Brm, SWI/SNF-like_EPAFa, WINAC, SWI/SNF-like EPAFB, bBAF	#

rSNP ID—a reference SNP identification number as in dbSNP; ENCODE protein—official symbol for DNA/chromatin binding protein tracked in ENCODE; and target genes—official symbols for genes with a ChIP-seq peak for a specific protein in promoter. Identifiers via EpiFactors: HMW/HME—histone modification writers/erasers, respectively; CR—chromatin remodelers; complex name—the protein acts as a component of designated protein complex if recognized in database; PMID—identifier for PubMed search for an associated publication; and #—a lack of information in the database.

## Data Availability

The data supporting the results reported in the paper can be accessed by request.

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
