# Peer review of "Exploring the Genetic Predisposition to Epigenetic Changes in Alzheimer’s Disease"

_ijms, 2023, doi:10.3390/ijms24097955_

Round 1
Reviewer 1 Report
In this study, the authors extracted regulatory SNPs (rSNPs) utilizing a dataset obtained from brains of Alzheimer’s disease (AD) and control cases published previously. These SNPs were selected by focusing on allele-specific binding data and allele-specific expression data obtained from ChIP-seq and RNA-seq, respectively, from the same individuals. Then, 1563 rSNPs were tested for their overlaps with the promoter of DEGs in AD, known epigenetic regulators, and transcription factors. The authors found a number of rSNPs that reside in the promoter region of these genes.
While the authors’ strategy focusing on the allele-specific binding and expression is interesting, it is unclear whether the rSNPs are causally or functionally associated with the observed changes in transcriptome and epigenome as well as AD etiology. I have several concerns as follows.
1. Although the authors cited their previous publication (ref 110), it is difficult to follow how they analyzed. In ref 110, they built alternative genomes, based on different data sets. This point is totally unclear in the current study. In addition, Supplementary Table 1 contains more than 1563 columns (the number of rSNPs). This may be because some rSNPs are associated with the allele specific expression of more than one gene. Unless clearly stated, this kind of discrepancy is confusing. Description of methodology should be refined.
2. Reproducibility of rSNP in different individuals is unclear. It should be evaluated whether the rSNPs show consistent effects (or associations) among multiple persons in the same data set or other data sets. For example, is there any difference in the expression level of ARID1B according to the genotype of rs228191 (or the expression level of other genes according to their cognate rSNP)? Supplementary Table 1 indicates most rSNP is derived from single person. Without such confirmation, functionality of rSNP is not convincing. This point is also related to the novelty of this study. As the importance of epigenome-related factors has been already highlighted in the original study (ref 32), the novelty of this study depends on the validity of rSNP.
3. Are there any overlaps between rSNPs and previously known DNA methylation QTLs (meQTLs)?
4. It seems that Supplementary Table 3 is a wrong one, distinct from what described in the text (page 6, line 189).
5. Regarding Figure 4, the author described that ARID1B binds to the promoter of 6 genes whose expression levels were predominantly elevated in AD. However, they also stated that all 6 genes contain rSNPs in their promoter regions. Which is important?
Author Response
Dear Editors, dear Reviewers,
We are very grateful for your interest and for comments to our manuscript. We have now revised the manuscript in accordance with your comments, and carefully proof-read it to minimize typographical, and grammatical errors. Here below is one-by-one response to your comments. Please, see the attachement for the improved manuscript text version.
Reviewer 1
Comment 1A. Although the authors cited their previous publication (ref 110), it is difficult to follow how they analyzed. In ref 110, they built alternative genomes, based on different data sets. This point is totally unclear in the current study.
Response. The alternative reference genomes were built for each individual under analysis. ChIP-seq and RNA-seq reads from all files of one human brain sample were pooled in order to increase the coverage. A couple of sentences have been added to bring clarity.
Comment 1B. In addition, Supplementary Table 1 contains more than 1563 columns (the number of rSNPs). This may be because some rSNPs are associated with the allele specific expression of more than one gene.
Response: The point is indeed that some rSNPs listed in the Supplementary Table 1 are associated with the allele specific expression of more than one target gene. The statement was added to the appropriate lines in Results section.
Comment 1C Unless clearly stated, this kind of discrepancy is confusing. Description of methodology should be refined.
Response: The raw data were initially processed as described by our previous publication, Korbolina et al. [doi:10.1002/humu.23425]. In brief, the raw data were preprocessed with respect to their quality and the coverage of heterozygous positions. Next, two consecutive binomial tests were applied, first- to RNA-seq, then – to ChIP-seq data in order to determine allele-specific events. Please, see details in Figure 1 and Methods section.
Comment 2. Reproducibility of rSNP in different individuals is unclear. It should be evaluated whether the rSNPs show consistent effects (or associations) among multiple persons in the same data set or other data sets. For example, is there any difference in the expression level of ARID1B according to the genotype of rs228191 (or the expression level of other genes according to their cognate rSNP)? Supplementary Table 1 indicates most rSNP is derived from single person. Without such confirmation, functionality of rSNP is not convincing. This point is also related to the novelty of this study. As the importance of epigenome-related factors has been already highlighted in the original study (ref 32), the novelty of this study depends on the validity of rSNP.
Response: As given in Discussion section, an important specific feature when using ASB and ASE for the detection of potential rSNPs is that the allele-specific events are identified and analyzed for each individual separately and thus - on the same background. This makes it possible to obtain the reliable data on functionality of the rSNP variant when studying even data obtained for a single person [Harvey, C.T. et al., QuASAR: quantitative allele-specific analysis of reads. Bioinformatics 2015, 31, 1235–1242, doi:10.1093/bioinformatics/btu802]. The reproducibility faces another problem of sample size: not all positions can be found in heterozygous state and thus analyzed in several individuals within one dataset. An increase in the sample size is needed for involving a larger number of the SNPs in a heterozygous state in the analysis.
The analysis of expression level of any target gene according to the rSNP genotype requires larger sample size as well because one needs to form the groups of several individuals of different homozygous and heterozygous genotypes before analyzing the differences in target gene expression. We have applied such approach successfully to link the identified rSNPs with the molecular phenotypes in our previous study (Novel functional variants at the GWAS-implicated loci might confer risk to major depressive disorder, bipolar affective disorder and schizophrenia, https://www.ncbi.nlm.nih.gov/pmc/articles/PMC5998904/).
Notice that the given data (GSE153875) suggest the reproducibility of allele-specific changes in protein binding and expression of target gene(s) if rSNP was found in ChIP-seq data for two or more people from the sample. The R plot below gives some details.
The bar plot shows that consistent allele-specific binding effects were found among multiple persons in GSE153875 dataset if the certain heterozygous position identified. It visualizes five summary statistics (the median as logarithm of fold change (log FC) in ChIP-seq data, two hinges and two whiskers) along the y-axis. The FC represents the ratio of the alternative allele to the reference allele in sequencing data. The rSNP identifiers (IDs) are given along the x-axis. The dots are the outliers, the dashes present the rSNP which were identified for the single person only.
Comment 3. Are there any overlaps between rSNPs and previously known DNA methylation QTLs (meQTLs)?
Response: In our study we did not aimed to overlap our regulatory genetic variants with the loci that influence DNA methylation variation across the genome as Nativio et collegues mainly focused on histone modification profiles affecting AD pathways.
We should notice that, first it would require additional data (importantly the data obtained for other samples) to implement in the analysis because the authors originally reported only the sequencing data for several nano-hmc-seal libraries. Secondly, there are certain difficulties to analyze allele-specific events when based on the data on genome methylation by any method, both Illumina methylation arrays and whole genome bisulfite sequencing (WGBS).
Comment 4. It seems that Supplementary Table 3 is a wrong one, distinct from what described in the text (page 6, line 189).
Response: This was checked up.
Comment 5. Regarding Figure 4, the author described that ARID1B binds to the promoter of 6 genes whose expression levels were predominantly elevated in AD. However, they also stated that all 6 genes contain rSNPs in their promoter regions. Which is important?
Response. We suppose that all identified regulatory features may be of importance in AD.

Reviewer 2 Report
Review of a manuscript “Exploring the Genetic Predisposition to Epigenetic Changes in Alzheimer's Disease” by LO Bryzgalov and coauthors submitted to IJMS.
Alzheimer's disease is the most prevalent neurodegenerative disease of the elderly, for which there is no efficient treatment changing the course of the disorder, so further studies are necessary to better understand molecular mechanism and find therapeutic targets. The authors investigated epigenetic mechanisms involved in Alzheimer's disease and its role in the disease etiology. This is an important area of biomedical investigation, and the manuscript contains new significant results which will be important for the readership of IJMS. The following corrections and additions should be made.
Abstract
Lines 12 and 14. The authors use twice the term “epigenetic landscape” close to each other. It would be beneficial to change one of them on another similar term, like “epigenetic panorama” or “epigenetic architecture” or “profile”
Line 16. “To this end, the available data (GSE153875} the authors should add more detailed information for better understanding, for example, GEO (gene expression omnibus) multi-omics approach.
Introduction
Lines 52-54
“The years of GWAS efforts and later, a meta-analyses of GWAS identified over 30 independent replicable LOAD genetic risk variants in addition to the APOE alleles [13– 15], including the SNPs in or close to CR1, BIN1, MEF2C, PICALM, SORL1, CLU, ABCA7, and CD33 genes.”
This is a long and clumsy sentence which should be split into two and presented in an easier to understand way.
Lines 70-72 “…histone modification patterns that alter the gene expression at a transcriptional level by upregulation, downregulation, or silencing [28].” After this sentence the authors should add a recent review on the role of epigenetic mechanisms in neurodegenerative diseases: ”alpha-Synuclein and mechanisms of epigenetic regulation. Brain Sciences, 2023, 13, 150. https://doi.org/10.3390/”.
Results
Lines 127- 128. “As a result, we identified 42 rSNPs in the promoters of 36 upregulated AD-related genes and 18 rSNPs in the promoters of 15 downregulated AD-related genes” The authors should explain how promoter region is defined in these analysis. How long, how far from the initiation start?
Line 159: ”…was subsequently recognized by the maintenance DNA methyltransferase DNMT1 [52]”
The authors should add maintenance DNA methyltransferase are essential for mammalian development.
Lipne 173-175:” The protein product of TRRAP …that lacks the enzymatic activity.” The authors should mention that TRRAP recruits HAT complexes to chromatin.”
Line 192: ”None of these 132 rSNPs were listed in any GWAS when related to AD [69].” Should be corrected as follows:” None of these 132 rSNPs were listed in any GWAS related to AD [69].”
Lines 238-239. Figure 3. Schematic representation of the analyzed regulatory interactions. T is unclear what does it mean the distance between -1000 bp and -1000 bp.
Lines 293-294: “Summing up, we have almost doubled the list of epigenetic regulators involved in AD as compared to the search for rSNPs in the promoter regions of AD-related genes here.” The sense of this sentence is not clear. The authors should rewrite it in easier to understand way.
Discussion
Line 407: ”The search for the rSNPs in promoters of the genes coding for transcription regulators for the genes in the distinguished group of AD-related genes was more effective.” The use of “genes” three times in one sentence makes the sentence awkward. It should be rewritten.
Overall, the manuscript contains important new results
Author Response
Dear Editors, dear Reviewers,
We are very grateful for your interest and for comments to our manuscript. We have now revised the manuscript in accordance with your comments, and carefully proof-read it to minimize typographical, and grammatical errors. Here below is one-by-one response to your comments. Please, see the attachement for the improved manuscript text version.
Comments to Abstract:
Lines 12 and 14. The authors use twice the term “epigenetic landscape” close to each other. It would be beneficial to change one of them on another similar term, like “epigenetic panorama” or “epigenetic architecture” or “profile”
Response: The term ‘epigenetic architecture’ was used in line 14, Abstract, with the respect to your comment.
Line 16. “To this end, the available data (GSE153875} the authors should add more detailed information for better understanding, for example, GEO (gene expression omnibus) multi-omics approach.
Response: The appropriate additions were made in the corresponding line.
Comments to Introduction:
Lines 52-54
“The years of GWAS efforts and later, a meta-analyses of GWAS identified over 30 independent replicable LOAD genetic risk variants in addition to the APOE alleles [13– 15], including the SNPs in or close to CR1, BIN1, MEF2C, PICALM, SORL1, CLU, ABCA7, and CD33 genes.”
This is a long and clumsy sentence which should be split into two and presented in an easier to understand way.
Response: We tried to present the information in a more understandable way.
Lines 70-72 “…histone modification patterns that alter the gene expression at a transcriptional level by upregulation, downregulation, or silencing [28].” After this sentence the authors should add a recent review on the role of epigenetic mechanisms in neurodegenerative diseases: ”alpha-Synuclein and mechanisms of epigenetic regulation. Brain Sciences, 2023, 13, 150. https://doi.org/10.3390/”.
Response: The additions were made in the text and the References section.
Comments to Results:
Lines 127- 128. “As a result, we identified 42 rSNPs in the promoters of 36 upregulated AD-related genes and 18 rSNPs in the promoters of 15 downregulated AD-related genes” The authors should explain how promoter region is defined in these analysis. How long, how far from the initiation start?
Response: The 4.5 section of Methods (Assignment of Gene Promoter Regions) explains the point: ‘The promoter regions were analyzed within 1 Mb windows in either direction from the TSSs.’
Line 159: ”…was subsequently recognized by the maintenance DNA methyltransferase DNMT1 [52]”
The authors should add maintenance DNA methyltransferase are essential for mammalian development.
Response: The additions were made in the text as follows: ‘…to either H3K9me2/3 or hemi-methylated CpG regions [53]. Note that DNMT1 methyltransferase and UHRF1 as its co-factor are the primal players in maintenance of imprinted methylation during the early development in mammals [ 10.1146/annurev-biochem-103019-102815, doi.org/10.3389/fcell.2020.629068].’
Line 173-175:” The protein product of TRRAP …that lacks the enzymatic activity.” The authors should mention that TRRAP recruits HAT complexes to chromatin.”
Response: The next sentence has already given the mention: ‘…The protein product of TRRAP …that lacks the enzymatic activity. TRRAP is an essential component of many histone acetyltransferase (HAT) complexes, which recruits the complexes to chromatin acting in transcription and DNA repair…’
Line 192: ”None of these 132 rSNPs were listed in any GWAS when related to AD [69].” Should be corrected as follows:” None of these 132 rSNPs were listed in any GWAS related to AD [69].”
Response: The changes were made with respect to the comment.
Lines 238-239. Figure 3. Schematic representation of the analyzed regulatory interactions. T is unclear what does it mean the distance between -1000 bp and -1000 bp.
Response: The distance between -1000 bp and +1000 bp fits the region that was considered the promoter region in the study. Please, see the section 4.5 in Methods. Two positions that were marked ‘-1000 bp’ on the Figure 3 limit the analyzed promoter regions on the left (we have now added the detail to the legend of the Figure 3). The point was to show that the identified rSNP and the ENCODe ChIP-seq peak are located within gene promoters. The positions ‘+1000bp’ were not shown on the Figure in order to simplify the drawing in details.
Lines 293-294: “Summing up, we have almost doubled the list of epigenetic regulators involved in AD as compared to the search for rSNPs in the promoter regions of AD-related genes here.” The sense of this sentence is not clear. The authors should rewrite it in easier to understand way.
Response: The appropriate corrections were made in the text as follows: ‘..Summing up, in the previous 2.2 section we gave a description of seven epigenetic regulators harboring rSNPs in their promoters, deregulated in AD and likely involved in pathogenic mechanisms. Here we identified eleven more genes (with totally fifteen more rSNPs in their promoters) that may add to epigenetic panorama in AD.’
Comments to Discussion:
Line 407: ”The search for the rSNPs in promoters of the genes coding for transcription regulators for the genes in the distinguished group of AD-related genes was more effective.” The use of “genes” three times in one sentence makes the sentence awkward. It should be rewritten.
Response: The appropriate corrections were made in the text as follows: ..’The search for the rSNPs in promoters of transcription regulators for the distinguished group of AD-related genes was more effective.’..

Reviewer 3 Report
The amnuscript "Exploring the Genetic Predisposition to Epigenetic Changes in 2 Alzheimer's Disease" analyzes SNPs thata can affect epigenetic mechanisms in AD. In the study authors use publicly available sequencing data. The topic of the articla is interesting and the article is well written. However I have some concerns and poits to be improved:
1.- Results section also include information that should be included as discussion
2.- Authors should explain deeper yhe methodology and criteria used for the search of SNPS
3.- An explanation of writers/readers/erasers machinary in the introduction would be useful
4.- The study is only based on public database information and is based in omics technology. The addition of sections with limitations of the study and the main conclusions would be useful for the readers.
Author Response
Dear Editors, dear Reviewers,
We are very grateful for your interest and for comments to our manuscript. We have now revised the manuscript in accordance with your comments, and carefully proof-read it to minimize typographical, and grammatical errors. Here below is one-by-one response to your comments. Please, see the attachement for the improved manuscript text version.
Reviewer 3
Comments and Suggestions for Authors
The manuscript "Exploring the Genetic Predisposition to Epigenetic Changes in Alzheimer's Disease" analyzes SNPs thata can affect epigenetic mechanisms in AD. In the study authors use publicly available sequencing data. The topic of the article is interesting and the article is well written. However I have some concerns and poits to be improved:
Comment 1.- Results section also include information that should be included as discussion
Response: Some minimal elements of the Discussion are indeed included in the Results section in order to facilitate both the discussion and understanding of the data.
Comment 2.- Authors should explain deeper the methodology and criteria used for the search of SNPS
Response: The raw data were initially processed as described by our previous publication, Korbolina et al. [doi:10.1002/humu.23425]. In brief, the raw data were preprocessed with respect to their quality and the coverage of heterozygous positions. Next, two consecutive binomial tests were applied, first- to RNA-seq, then – to ChIP-seq data in order to determine allele-specific events. Please, see details in Figure 1 and Methods section. Some appropriate changes were made in the text.
Note that the very close criteria were reported in several publications including MBASED: allele-specific expression detection in cancer tissues and cell lines (https://www.ncbi.nlm.nih.gov/pmc/articles/PMC4165366/), ASEP: Gene-based detection of allele-specific expression across individuals in a population by RNA sequencing (https://www.ncbi.nlm.nih.gov/pmc/articles/PMC7241832/#pgen.1008786.ref014).
Comment 3.- An explanation of writers/readers/erasers machinary in the introduction would be useful
Response: The appropriate changes were made in the Introduction including the relevant quotation.
Comment 4.- The study is only based on public database information and is based in omics technology. The addition of sections with limitations of the study and the main conclusions would be useful for the readers.
Response: Thanks a lot for your valuable comment. We made an addition in Discussion section with a special attention to existing limitations to the study.
